

# Analysis of rotational grazing management for sheep in mixed grassland

Zongyong Tong[1], Xianlin Dai[1], Yu Wang[1], Xianglin Li[1], Feng He[1] and Guomei Yin[2]

[1] Institute of Animal Sciences, Chinese Academy of Agricultural Sciences, Beijing, China
[2] Inner Mongolia Academy of Agricultural and Animal Husbandry Sciences, Hohhot, China

## ABSTRACT

Sown mixed grassland is rarely used for livestock raising and grazing; however, different forages can provide various nutrients for livestock, which may be beneficial to animal health and welfare. We established a sown mixed grassland and adopted a rotational grazing system, monitored the changes in aboveground biomass and sheep weights during the summer grazing period, measured the nutrients of forage by near-infrared spectroscopy, tested the contents of medium- and long-chain fatty acids by gas chromatography, and explored an efficient sheep fattening system that is suitable for agro-pastoral interlacing areas. The results showed that the maximum forage supply in a single grazing paddock was 4.6 kg DM/d, the highest dry matter intake (DMI) was 1.80 kg DM/ewe/d, the average daily weight gain (ADG) was 193.3 g, the DMI and ADG were significantly correlated ($P < 0.05$), and the average feed weight gain ratio (F/G) reached 8.02. The average crude protein and metabolizable energy intake by sheep were 286 g/ewe/d and 18.5 MJ/ewe/d respectively, and the n-6/n-3 ratio of polyunsaturated fatty acids in mutton was 2.84. The results indicated that the sheep fattening system had high feed conversion efficiency, could improve the yield and quality of sheep, and could be promoted in suitable regions.

## INTRODUCTION

The traditional methods of raising mutton sheep in northern China include pasture grazing in natural pastoral areas, house feeding in agricultural areas and semigrazing feeding in particular areas (*Chu, Hou & Jiang, 2022*). For a long time, the development of animal husbandry in grassland areas has been heavily dependent on natural grassland, which has led to long-term overgrazing, grassland degradation and desertification, resulting in an imbalance in grassland ecological and productive functions (*Bardgett et al., 2021*). However, in most agricultural areas, the total house-feeding system has been widely adopted, which has resulted in lower livestock activity, reduced mutton quality and more severe disease (*Liu et al., 2022*). Modern grassland and animal husbandry are new industrial patterns, a remarkable feature of which is the combination of planting and animal husbandry (*Hou*

Corresponding authors
Feng He, hefeng@caas.cn
Guomei Yin, gmynmg@126.com

& Zhang, 2018). Forage is the material basis for the development of animal husbandry, and more than 60–70% of the current fattening cost is feed (Becker, 2012). Therefore, the efficient utilization of forage is an important step in reducing production costs and improving productive efficiency and industrial benefits.

Sown mixed grassland is an herb community that is planted with fine herbage to obtain stable, high-yield and high-quality forage *via* artificial actions. It is generally composed of perennial legume (*e.g.*, *Medicago sativa* L. (alfalfa) and *Trifolium repens* (white clover)) and grass (*e.g.*, *Lolium perenne* (ryegrass), *Bromus inermis* (smooth brome) and *Leymus chinensis*) species; combinations of these species in different ecological niches could make full use of natural environmental resources and achieve nutritional balance. The typical example is the clover–ryegrass grassland under moist, fertile and intensely grazed conditions in New Zealand, Australia and other countries with similar conditions. It can not only compensate for the low yields of natural grasslands and effectively relieve the grazing pressure on grasslands, but also provide high-quality forage for livestock (*Zhao et al., 2020*). In recent years, with the development of animal husbandry, how to efficiently utilize sown mixed grasslands for grazing has become a research hotspot. Previous studies have shown that sown mixed grasslands can obtain more stable, high-yield and high-quality forage than can natural grasslands by planting high-quality forage and adopting modern management measures (*Wang et al., 2021*).

The existing grazing systems mainly include continuous grazing and rotational grazing; however, continuous grazing, in which livestock have unrestricted access to the pasture area, often results in overgrazing and pasture degradation. The rotational grazing system for moving livestock from paddock to paddock is based on available forage, paddock size and livestock growth goals. In contrast to continual grazing, a rotational grazing system can ensure that the grassland has enough time to rest, prevent grassland degradation, improve forage yields and utilization rates, reduce the infection rate of livestock diseases, and achieve a balance between productive and ecological benefits (*Liu et al., 2017*). Rotational grazing systems can improve grassland conditions; consider both economic development and ecological environmental protection; achieve the harmonious coexistence of human settlements, grasslands and livestock; and ensure the healthy development of grassland ecosystems (*Ren, 2012*; *Teague & Kreuter, 2020*). The number of paddocks and rotational management depends on many factors, such as pasture productivity, climate, and livestock species. How to efficiently utilize grasslands and improve livestock production efficiency through rotational grazing management is an important issue.

Semiagricultural and semipastoral areas (also known as agro-pastoral interleaved areas) are the transition areas between agricultural planting areas and animal husbandry areas, as well as the interface where the two major production systems converge, where the available grazing area is limited, and the quality of cultivated land is infertile; however, these areas are suitable for the development of modern grassland animal husbandry. The rational and efficient utilization of grasslands for fattening is one of the main problems faced by these areas.

In this study, we established and planted artificial legume/grass mixed-sowing grassland and carried out a rotational grazing experiment and sheep fattening system to study

the relationship between grassland biomass or nutrient composition and the grazing of sheep, as well as the production performance of sheep through the system, to evaluate the mixed-sowing grassland fattening system and balance the ecological and productive functions in relevant areas.

## MATERIALS & METHODS

### Study area

The experimental site was located in Langfang city (39°35′44″N, 116°34′60″E), Hebei Province, China. Over the past 10 years, the mean annual rainfall at the study site was 554 mm, with a peak in July, and the annual average temperature was 11.9 °C. The average annual frost-free period is 183 days, and the average annual sunshine duration is 2660 h. The soil is a sandy loam in the Chinese classification system and a Calcic-Orthic Aridisol in the US system, with 0.37 g/kg organic matter, 31.2 mg/kg alkali-hydrolyzed nitrogen, 0.87 mg/kg available phosphorus, 85.1 mg/kg available potassium and a pH of 7.8.

### Experimental livestock

Forty-nine healthy male Dumeng dihybrid sheep aged 4–5 months were selected from Inner Mongolia Sino-Sheep Science and Technology Co., Ltd. The body weight of each sheep was 30 ± 2.0 kg. The research was fully approved by the Experimental Animal Welfare and Ethics Committee of the Institute of Animal Science, Chinese Academy of Agricultural Sciences (IAS2021-247). All the sheep grazed freely in the mixed grassland during the day, drank water, were provided supplemented feed and rested in a house at night. All procedures were conducted in accordance with the "Guiding Principles in the Care and Use of Animals" (China).

### Mixed grassland

The mixed grassland was sown in September 2016; the sown species consisted of a mixture of legume and grasses. The legume was alfalfa, the sowing rate of which was 30 kg/ha, accounting for 75%, and the grasses were *Festuca arundinacea* (tall fescue) and *Dactylis glomerata* (orchardgrass), the sowing rate of which was 5 kg/ha, accounting for 12.5% of the total weight. After the seeds were mixed evenly, the grass was sown in lines with a depth of 2 cm and row spacing of 15 cm. In 2021, the quadrat statistics of each 1 × 1 m plot showed that the dry matter ratio of alfalfa accounted for 80–90%, tall fescue and orchardgrass accounted for 5% and weeds accounted for 5–15% of the whole grassland community.

### Experimental design

The enclosed grazing land totaled 1.6 ha and was equally divided into six paddocks; each paddock was 0.27 ha, and P1 to P6 were marked successively (Fig. 1). Adaptation training was adopted for 5 days before starting the grazing test to prevent distention disease. Sufficient gramineal hay was provided before grazing, with 1 h of grazing on the first day (0.5 h in the morning, 0.5 h in the afternoon) in the main pasture path, followed by an increase of 1 h the next day, and free grazing on the 5th day. The rotational grazing
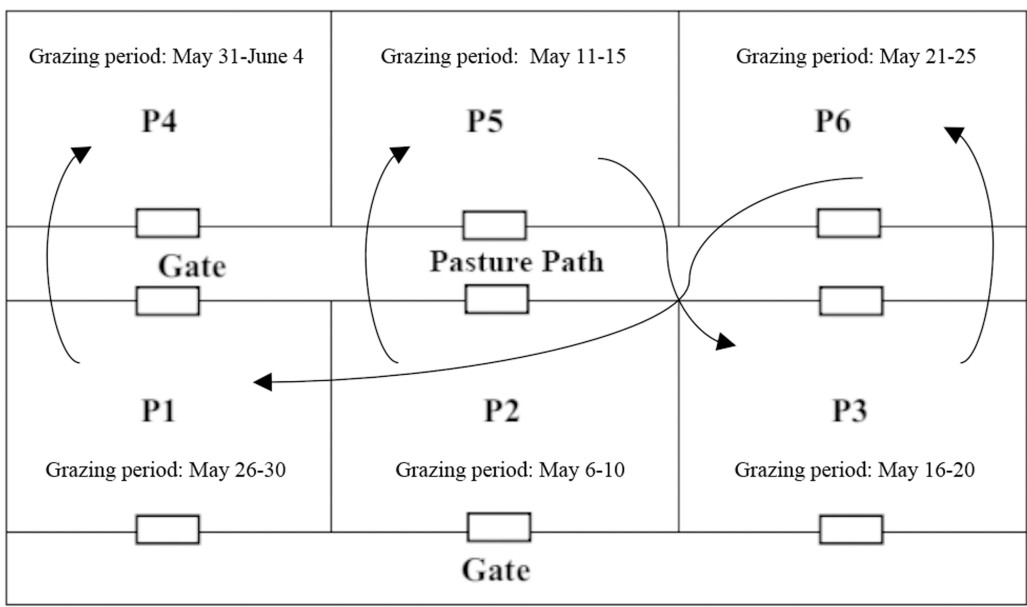

**Figure 1** Rotational grazing experiment area.

experiment began when the aboveground biomass in a paddock reached 900 kg DM/ha (*Wang et al., 2022*).

The rotational grazing test was conducted on May 6, 2021; the stocking rate was 30 ewes/ha, the rotation time per paddock was 5 days, the stocking period per cycle was 30 days, and five stocking cycles were established per year. All the sheep were freely grazed from paddock P2, followed by P5, P3, P6, P1 and P4, for 5 days in each paddock, and the grazing time ranged from 6:00 to 18:00 for 12 h every day. At 19:00, each sheep was supplemented with 100 g of crushed corn kernels. The nutritional composition of the corn kernels is shown in Table 1. The grazing paddock was equipped with automatic drinking devices and lick bricks to supply water and minerals.

## Aboveground biomass

Aboveground plants (stubble height <5 cm) were collected in 1 × 1 m quadrats from the grazing paddock with three repetitions on the day when the sheep pregrazing and postgrazing. After the plants were weighed, they were dried in a forced-air oven (Model No. XMTD-8222, JingHong, Shanghai, China) at 65 °C to a constant weight.

## Dry matter intake

The dry matter contents of plant samples in each grazing area were measured after drying, and the average apparent daily dry matter intake (DMI) per sheep in each grazing paddock for 5 days was calculated by the following equation:

$$DMI = \frac{DM1 - DM2}{49 \times 5}$$

DM1: Dry matter pregrazing
DM2: Dry matter postgrazing.
**Table 1  Nutrient levels of corn (dried matter basis).**

| Feed | Dry matter/% | Crude protein/% | Ether extract/% | Crude fiber/% | Metabolizable energy/(MJ/kg) |
|---|---|---|---|---|---|
| Corn grain | 88.0 | 7.5 | 3.0 | 10.0 | 13.4 |

## Plant nutrition

The dry weight samples from each grazing area were ground with a grinder. The powder was filtered through a 1.0 mm filter and then analyzed by a FOSS NIRS DS2500 near-infrared spectrum analyzer (FOSS, Hilleroed, Denmark) for crude protein content (CP, %) and acid detergent fiber (ADF, %) according to previously described methods (*Wood et al., 2018*). NIRS measurements were calibrated by the partial least squares (PLS) method. The typical standard error of calibration (SEC), standard error of prediction (SEP), and standard error of cross validation (SECV) were 1.64, 1.83, and 1.66, respectively, for ADF and 0.83, 1.01, and 0.84, respectively, for CP.

The metabolizable energy (ME) was calculated by the following equation (*Basarab et al., 2007*):

$$TDN = 88.9 - 0.779 \times ADF \times 100\%$$
$$ME(MJ/kg) = TDN \times 0.01 \times 4.4 \times 4.185.$$

## Sheep weight

Each time the sheep entered and left the grazing area, every sheep was weighed with a weighing scale (Yikangnong Science and Technology Development Co., Ltd., Beijing, China) installed at the gate of the paddock at the same times each day.

## Fatty acids

At the conclusion of grazing, all the sheep were euthanized. The euthanasia procedures were in accordance with the *American Veterinary Medical Association (2020)*. Euthanasia was performed using an injectable anesthetic (pentobarbital) overdose (>100 mg/kg), and the animals were monitored until a lack of heartbeats was noted for >60 s prior to tissue harvesting. Six sheep with similar weights were selected for mutton fatty acid analysis. Two hundred grams of longissimus dorsi muscle samples were collected immediately after slaughtering, transported to the laboratory immediately in fresh-keeping bags in an ice-box, and stored at −80 °C.

Each muscle sample was crushed, and 1.5 ml of n-hexane methyl undecanoate was added to 0.5 g of muscle sample as the internal standard. Fatty acids were reacted with potassium hydroxide/methanol in a 90 °C water bath for 30 min after shaking and mixing. After centrifugation, the supernatant was passed through a 0.22 μm organic filter membrane and used as the loading liquid, and fatty acid methyl esters were identified and quantified by an Agilent 7890A gas phase mass spectrometer (GC/MS) (Agilent Technologies, Santa Clara, CA, USA). The detection and quantification limits were 5 mg/kg (ppm), and all muscle samples were prepared in duplicate to check the repeatability.

## Data analysis

Microsoft Excel 2010 (Microsoft Corp., Redmond, WA, USA) was used to record the experimental data, SPSS 19.0 (SPSS, Chicago, IL, USA) software was used for variance and regression analyses, multivariate ANOVA (MANOVA) was used to test the significance of differences among paddocks, and the results were recorded as the mean $\pm$ standard error. $P < 0.05$ was considered to indicate a significant difference.

## RESULTS

### Supplied biomass of each paddock

The biomass of the mixed grassland pregrazing in each paddock, and the forage supply for each sheep in the 49 flocks were presented in Table 2. The forage supplies in the P6, P1 and P4 paddocks with later grazing times were greater than those in the first three paddocks, reaching a maximum of 4.6 kg DM/ewe/d in P4, while that in the P2 region was only 3.21 kg DM/ewe/d, and those in the P2 and P3 paddocks was significantly lower than those in the other paddocks ($P < 0.05$). The average pregrazing biomass in each paddock was 3,666.3 kg DM/ha, and the forage available to each sheep was 3.99 kg DM/ewe/d.

### Dry matter intake in the grazing paddocks

The aboveground biomass deceased after the sheep grazed in each paddock, and the DMI for every sheep is given in Table S1. Statistical analysis revealed that there were significant differences in the DMI of the sheep in each paddock ($P < 0.05$). The highest DMIs in P3 and P4 were 1.80 and 1.79 kg DM/ewe/d, respectively, while the DMIs in P5 and P6 were only 1.11 and 1.17 kg DM/ewe/d, respectively, and the average DMI was 1.46 kg DM/ewe/d (Fig. 2).

### Crude protein and metabolizable energy intake in each grazing paddock

When grazing began in May, the grassland conditions of every paddock were nearly the same. When the experimental period began in July, every paddock was grazed twice. Aboveground vegetation was sampled every day in each paddock during the grazing period, CP and ADF were measured by a near-infrared detector (Table S2), and ME was calculated. Then, CP and ME intakes were calculated according to the DMI. Each index value was the average value during grazing (Table 3).

The CP content in the P6 paddock was 20.28%, the highest of all paddocks, and that in the P4 paddock was the lowest, at only 18.12%. There was no significant difference in CP content among the paddocks ($P > 0.05$). However, due to the greater DMIs in P3 and P4, the CP intakes in the two paddocks were significantly greater than those in the other paddocks ($P < 0.05$). P3 had the highest CP intake (345.19 kg/ewe/d), while P5 had the lowest CP intake (208.88 g/ewe/d), with an average CP intake of 278.83 g/ewe/d.

The highest ME intakes were in P2 and P5, both of which reached 12.11 MJ/kg, and the average ME intake was 11.77 MJ/kg. The ME intake was calculated according to the feed intake in each paddock. The highest ME intake in P4 was 21.23 MJ/ewe/d, and the average ME intake in all paddocks was 17.15 MJ/ewe/d.

Tong et al. (2024), *PeerJ*, DOI 10.7717/peerj.17453

**Table 2  Pre-grazing biomass and supplied biomass of each paddock.**

| Paddock no. | P2 | P5 | P3 | P6 | P1 | P4 | Average | P value |
|---|---|---|---|---|---|---|---|---|
| Pregrazing/ (kg DM/ha) | 2,952.5 ± 208.7[c] | 3,622 ± 163.9[ab] | 3,132 ± 34.8[bc] | 3,968 ± 259.4[a] | 4,097.8 ± 129.7[a] | 4,225.8 ± 298.4[a] | 3,666.3 ± 182.5 | 0.01 |
| Supplied/ (kg DM/ewe/d) | 3.21 ± 0.23[c] | 3.94 ± 0.18[ab] | 3.41 ± 0.04[bc] | 4.32 ± 0.28[a] | 4.46 ± 0.14[a] | 4.60 ± 0.32[a] | 3.99 ± 0.20 | 0.01 |

**Notes.**
The significant difference between different treatments of the same trait is denoted as different lowercase letters in the same line.

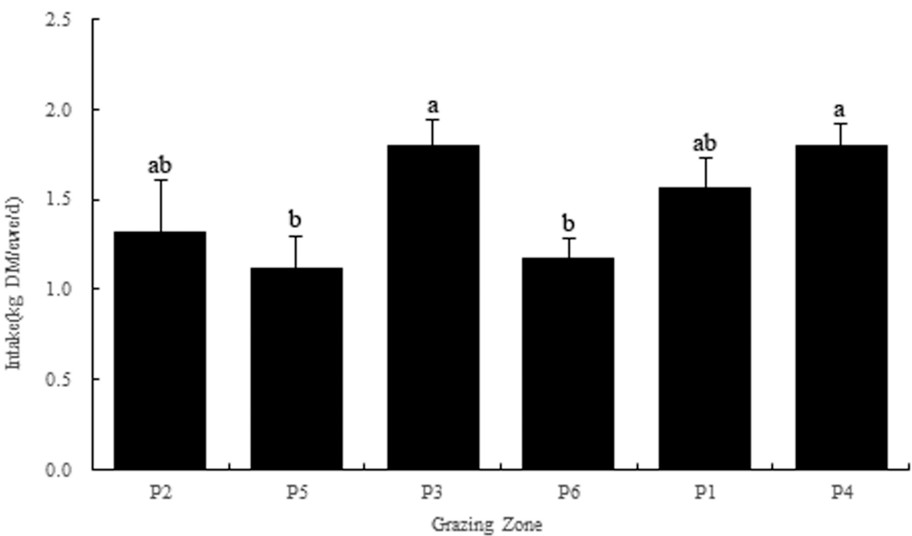

**Figure 2  Dry matter intake in each grazing paddock.** Different lowercase letters a, b and ab indicate significant differences at the 0.05 level.

**Table 3  Nutrient levels and intake of each grazing paddock (dried matter basis).**

| Paddock no. | CP, % | ADF, % | ME/(MJ/kg) | CP intake/(g/ewe/d) | ME intake/(MJ/ewe/d) |
|---|---|---|---|---|---|
| P2 | 20.19 ±0.71[a] | 29.73 ±0.42[c] | 12.11 ±0.11[a] | 266.53 ±9.24[cd] | 15.98 ±0.14[b] |
| P5 | 18.82 ±1.29[a] | 29.66 ±0.93[c] | 12.11 ±0.12[a] | 208.88 ±12.35[e] | 13.45 ±0.13[d] |
| P3 | 19.18 ±1.16[a] | 32.81 ±0.74[ab] | 11.66 ±0.11[bc] | 345.19 ±17.18[a] | 20.99 ±0.19[a] |
| P6 | 20.28 ±0.37[a] | 33.90 ±1.64[a] | 11.51 ±0.08[c] | 237.26 ±4.36[de] | 13.47 ±0.09[d] |
| P1 | 19.10 ±0.46[a] | 34.38 ±1.22[a] | 11.44 ±0.10[c] | 297.88 ±7.22[bc] | 17.85 ±0.15[b] |
| P4 | 18.12 ±0.68[a] | 31.45 ±0.77[bc] | 11.86 ±0.06[ab] | 324.27 ±12.25[ab] | 21.23 ±0.11[a] |
| Average | 19.28 | 32.06 | 11.77 | 278.83 | 17.15 |

**Notes.**
The significant difference between different treatments of the same trait is denoted as different lowercase letters in the same column.

Because mixed-sowing grasslands lack an energy supply, it is necessary to supplement 100 g of crushed corn every day. The CP content of the supplementary corn was 7.5 g/ewe/d, and the ME intake was 1.34 MJ/ewe/d. Therefore, the total CP intake of the meat sheep was approximately 286 g/ewe/d, and the total ME intake was 18.5 MJ/ewe/d.

## Weight gain and feed weight gain ratio under rotational grazing

Sixteen sheep with similar growth states and body weights were selected, and the sheep in each paddock were weighed before and after grazing (Table S3) to monitor the effect of grazing on sheep body weight. The average daily gain (ADG) reached 193.3 g/d, with the highest ADG being 268 g/d (Table 4). In P1 and P4, the ADGs were 226 g/d and 268 g/d, respectively, but in P2 and P5, the ADGs declined to 154 g/d and 138 g/d, respectively, and there was no significant difference in weight gain among the different paddocks ($P > 0.05$).

**Table 4  The ratio of feed and weight gain in each grazing paddock.**

| Paddock no. | Total DMI/(kg/ewe/d) | ADG/(g/ewe/d) | F/G |
|---|---|---|---|
| P2 | 1.41 ±0.29[ab] | 154 ±30[a] | 9.14 |
| P5 | 1.20 ±0.19[b] | 138 ±48[a] | 8.70 |
| P3 | 1.89 ±0.14[a] | 206 ±32[a] | 9.17 |
| P6 | 1.26 ±0.11[b] | 168 ±32[a] | 7.52 |
| P1 | 1.65 ±0.17[ab] | 226 ±35[a] | 7.32 |
| P4 | 1.88 ±0.12[a] | 268 ±58[a] | 7.03 |
| Average | 1.55 | 193.3 | 8.02 |

Notes.
The significant difference between different treatments of the same trait is denoted as different lowercase letters in the same column.

The use of mixed grasslands and rotational grazing systems is important for improving the forage conversion rate. The average ratio of feed to weight gain (F/G) is an important index for evaluating forage conversion efficiency (FCE). One hundred grams of crushed corn was added daily, the total DMI = DMI of grassland + DMI of corn, the dry matter content of crushed corn was 88%, and the DMI of corn was approximately 0.09 kg DM/ewe/d. The total DMI and F/G values of each sheep in each paddock were calculated (Table 4). The average F/G was 8.02; that is, approximately 8 kg of feed could be converted into 1 kg of sheep weight in this fattening system.

## Relationships between the forage supply, total DMI and ADG of sheep

Sheep obtain nutrition and energy from feed and convert it into their own matter and energy, which is significantly reflected in changes in body weight. Therefore, the forage supply, total DMI and ADG could be correlated, and the correlation analysis of the three indices is shown in Table 5.

The daily forage supply was 3.99 kg DM/ewe/d, while the total DMI was 1.55 kg DM/ewe/d. When the forage supply was greater than the demand of the sheep, the forage supply was not significantly correlated with the DMI or ADG, but the DMI and ADG were significantly correlated ($P < 0.05$).

The DMI and ADG in each paddock were analyzed by regression analysis. The total DMI and ADG conformed to a sigmoidal equation model, and nonlinear curve fitting was conducted with a fair fitting degree ($R^2 = 0.778$) (Fig. 3).

## Fatty acid contents of mutton in the mixed grassland grazing system

The various fatty acids (FAs) in the longissimus dorsi muscles of six sheep were measured by GC/MS analyses (Table 6). The recovery levels of fatty acids ranged from 82–90% after extraction with potassium hydroxide/methanol, and the coefficient of variation (CV) of precision for each fatty acid was less than 8%. The total FA content of the mutton was 17,433.33 mg/kg, the saturated fatty acid (SFA) content was 10,170 mg/kg, the monounsaturated fatty acid (MUFA) content was 4,131.67 mg/kg, and the polyunsaturated fatty acid (PUFA) content was 3,131.67 mg/kg. The contents of palmitic and stearic acids accounting for most of the SFAs, were 5,020 mg/kg and 3,801.67 mg/kg, respectively. The

**Table 5   Correlation analysis of forage supplies, dry matter intake and weight gain per day.**

| | Index | Forage supply | Total DMI | ADG |
|---|---|---|---|---|
| Forage supply | Pearson Correlation | 1 | 0.094 | 0.558 |
| | Sig. (two-tailed) | – | 0.859 | 0.250 |
| Total DMI | Pearson Correlation | 0.094 | 1 | 0.865[*] |
| | Sig. (two-tailed) | 0.859 | – | 0.026 |
| ADG | Pearson Correlation | 0.558 | 0.865[*] | 1 |
| | Sig. (two-tailed) | 0.250 | 0.026 | – |

**Notes.**
*A significant difference at the 0.05 level (two-tailed).

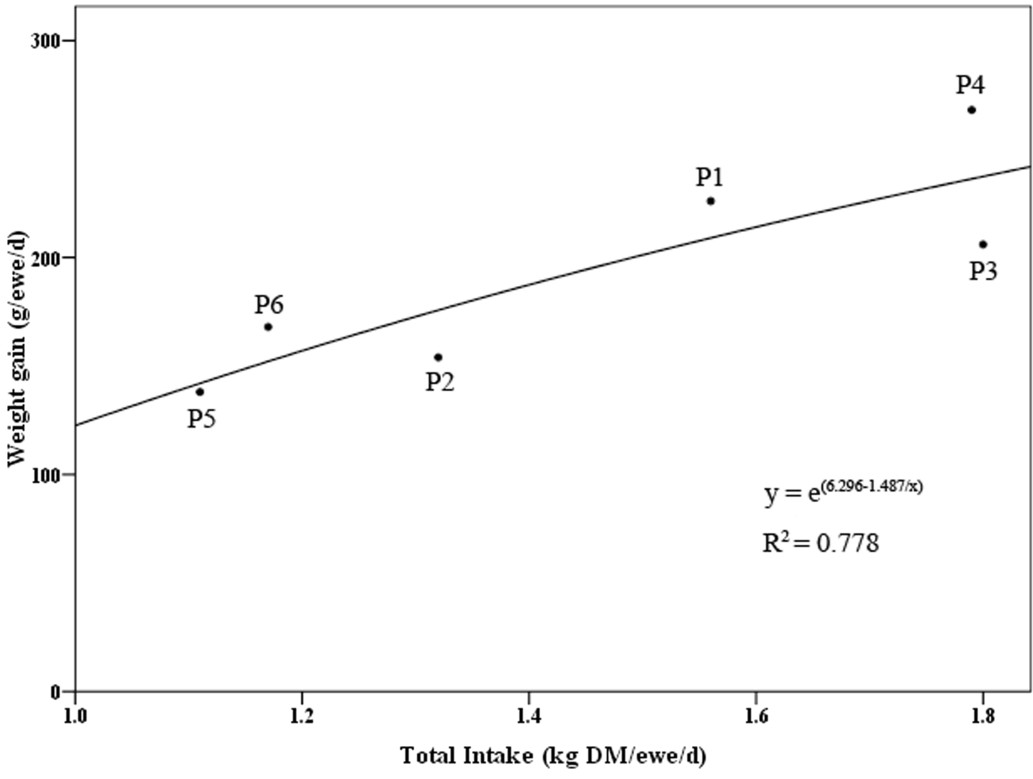

**Figure 3   Fitting curve of sheep intake and weight gain per day.**

content of oleic acid in the MUFAs was 3,801.67 mg/kg, the content of n-6 in the PUFAs was 2,213.33 mg/kg, the content of n-3 was 780 mg/kg, and the ratio of n-6: n-3 was 2.84.

## DISCUSSION

### Nutrition and energy supply in the rotational grazing system

According to the regulations of the National Research Council (NRC) and the Chinese Agricultural Industry Standard (Feeding Standard for Meat Sheep, NY/T 816), a 40 kg group of sheep needs 1.4 kg DM/d, 18.40 MJ/d ME and 204 g/d CP to achieve a weight gain

**Table 6  Fatty acid contents in mutton.**

| Category | Fatty acids | Contents/(mg/kg) |
|---|---|---|
| SFA | Myristic acid | 513.33 ±68.55 |
|  | Pentadecanoic acid | 95 ±12.59 |
|  | Palmitic acid | 5020 ±681.15 |
|  | Heptadecanoic acid | 301.67 ±31.99 |
|  | Stearic acid | 4128.33 ±432.72 |
|  | Total SFA | 10170 ±1175 |
| MUFA | Palmitoleic acid | 263.33 ±37.48 |
|  | Oleic acid | 3801.67 ±530.80 |
|  | Cis-11-Eicosenoic acid | 40 ±5.17 |
|  | Total MUFA | 4131.67 ±574.02 |
| PUFA | Linoleic acid, n6 | 1598.33 ±75 |
|  | $\alpha$-Linolenic acid, n3 | 376.67 ±36.49 |
|  | cis-11,14-Eicosadienoic acid | 103.33 ±7.60 |
|  | cis-8,11,14-Eicosatrienoic acid, n6 | 70 ±4.47 |
|  | Arachidonic (ARA), n6 | 536.67 ±29.41 |
|  | Eicosapentaenoic acid (EPA), n3 | 333.33 ±16.06 |
|  | Docosahexaenoic Acid (DHA), n3 | 70 ±7.3 |
|  | Conjugated linoleic acid (CLA) | 35 ±6.19 |
|  | Total PUFA | 3131.67 ±120.34 |
|  | $n-6/n-3$ | 2.84 ±0.06 |
|  | P/S | 0.326 ±0.04 |
| FA | Total FA | 17433.33 ±1805.97 |

**Notes.**

Some fatty acids with lower contents are not listed, data are mean ± SE.

of 0.3 kg/d (*National Research Council, 2007*), which requires the feed to contain 14.6% CP and 13.14 MJ/kg ME. The CP content of the natural grassland in Xilin Gol, one of the best-preserved areas of natural grassland in China, was 11.48–16.41% (*Shi, 2012*), while the CP contents in the slope, flat and low-lying natural grasslands were approximately only 11%, 11.8% and 7%, respectively (*Fu, Wang & Li, 2022*). These results indicated that the CP content of the forage in natural grassland was not high enough to meet the nutritional requirements of livestock. Moreover, rotational management also resulted in 28% and 20% greater total herbage production, respectively, than traditional stocking (*Schons et al., 2021*). However, in our study, the mixed grassland mainly sown with alfalfa was rich in crude protein, the content of which was 19.28%, but the content of starch and other energy substances was less than 10%. To prevent bloating from high alfalfa consumption, sheep were adaptively trained to feed them grass hay and concentrate, and the grazing time was gradually extended (*Wang et al., 2023*; *Wang, Majak & McAllister, 2012*). Therefore, to prevent various diseases caused by nutritional imbalances, energy feeds, such as corn, are needed to achieve an energy and protein balance. In the system, the forage supply of each sheep was more than 1.55 kg/d DM, the CP intake was approximately 286 g/ewe/d, and the ME intake was 18.5 MJ/ewe/d, which fully met the feeding standards of meat sheep.

Therefore, the productivity of the artificial mixed grassland was much greater than that of natural grassland.

## Relationship between feed intake and daily gain under the rotational grazing system

In this experiment, stable forage and nutrient supplies were provided for sheep through rotational grazing in the demarcated area, and the ADG reached 193.3 g/d. The seasonal change in forage quality in mixed pastures also affected the feed intake and weight gain of the sheep. In the system, in May, the CP and ADF contents of the mixed-sown grassland were 20.92% and 31.83%, respectively. However, in August, the CP content decreased to 12.31%, the ADF content increased to 43.4%, and the forage quality decreased significantly. Similarly, the ADG of mutton sheep from May to July was 216 g/d, while that from July to September decreased to 153 g/d (*He et al., 2020*). In our experiment, the CP of the mixed-sown grassland also decreased from 20.19% to 18.12%, the ADF increased from 29.73% to 34.38% (from July to August), and the ADG of sheep was 193.3 g/d, which indicated that the forage quality of the mixed-sown grassland was directly related to the weight gain of the mutton sheep.

There was no significant difference between the DMI and forage supply of the mutton sheep, perhaps because the supply was much greater than the demand, and there was no correlation between the two factors. However, DMI and ADG were significantly correlated and presented a sigmoidal curve, and the F/G value of forage conversion was 8.02. In addition, the FCE under rotational stocking was 40% greater than that under traditional stocking (*Schons et al., 2021*). Previous studies have shown that the daily weight gain resulting from free grazing in natural grasslands is approximately 100 g/d (*Li, Zhou & Lin, 2005*). Under grazing conditions in a typical grassland, the conversion efficiency of grass and livestock was 8.73–11.32%, and the F/G was 8.8–11.5 (*Cong et al., 2017*). The ADG of house-fed sheep fed a granular TMR was 293.44 g/d, the DMI was 1,845.1 g/d, and the F/G was 6.29 (*Cheng et al., 2022*). Specifically, for other four-hoofed animals, the DMI of dairy cattle grazing in grass–birdsfoot trefoil mixed pastures could reach 4–4.4 kg/d, the FCE was approximately 0.12, and the F/G was 8.3 (*Greenland et al., 2023*). The FCE of suckler-bred steers in rotationally grazed perennial ryegrass-dominant pastures was approximately 1.2 (F/G = 9.1) (*Doyle et al., 2022*). A meta-analysis of performance in cattle grazing on tropical grasslands revealed that the global FCE was 0.11 (F/G = 9.1), and the FCE was enhanced by the level of ADG (*Boval, Edouard & Sauvant, 2015*). In conclusion, the ADG of sheep grazed on high-quality alfalfa mixed pasture was generally greater than those of sheep and cattle grazing in grasslands and lower than that of concentrate feeding in houses. Due to the high protein but low carbohydrate contents in forage, a certain amount of concentrate needs to be added to provide energy, and the forage conversion rate, as an index of F/G, is generally greater than that in natural grasslands. This value is lower than that of concentrated feed or other energy feeds, indicating that alfalfa mixed-grassland grazing has great application potential and can be further improved through optimizing the forage ratio and management.

## Fatty acid content in mutton

The fatty acid content of mutton is mainly determined by the type of diet. By comparing the fatty acid contents of Sunit sheep under the two feeding methods, it was found that the stearic acid, trans-oleic acid, trans-linoleic acid (n-6), arachidonic acid (n-6) and docosahexaenoic acid (n-3) contents of the grazing sheep were significantly greater than those of the house-feeding sheep ($P < 0.05$) (*Li et al., 2019*). When sheep were fed hay feed, the $n-6/n-3$ ratio was 1.28, while when lambs were fed concentrate containing a high content of n-6 fatty acids such as linoleic and arachidonic acids, the $n-6/n-3$ ratio increased to 7.11 (*Demirel et al., 2006*). Similar findings were also observed in this study. After grass grazing, the $n-6/n-3$ ratio of mutton was 2.84. Studies have shown that PUFAs not only are related to mutton flavor but also have many benefits to human health (*An et al., 2018*). The United Nations Food and Agriculture Organization recommends that the $n-6/n-3$ ratio be $\leq 4$ (*Simopoulos, 2008*), and grazing or grass-fed mutton meets this standard.

As saturated fatty acids (SFAs) can increase the content of LDL cholesterol in human blood, while polyunsaturated fatty acids (PUFAs) have the opposite effect, the ratio of the two types of fatty acids (P/S) is also an important index for measuring the nutritional value of meat. Nutritionists generally believe that the optimal P/S for human health is approximately 0.4 (*Hayes, 2002*). In previous studies, the P/S in mutton generally ranged from 0.10–0.26 (*Banskalieva, Sahlu & Goetsch, 2000*), while in this study, the P/S value of mutton was 0.326, which was close to the optimal value, indicating good nutritional value.

## CONCLUSIONS

In conclusion, the productivity of mixed grassland undergoing rotational management surpassed that of natural grassland, and in the rotational grazing system, the correlation between the DMI and ADG of sheep was determined with a sigmoidal equation model. The average F/G had a high FCE, and the $n-6/n-3$ ratio of polyunsaturated fatty acids in mutton was more beneficial to human health. The specific components of mixed grasslands that influence the fatty acid content and other quality traits of mutton through specific pathways require further investigation.

## ACKNOWLEDGEMENTS

We thank Bao Wei and Xiaoying Ma for their assistances with field data collection, and Lihong Miao for materials and accounting.

### Funding

This work was supported by the Inner Mongolia Science and Technology Project (NO. 2021GG0220 and 2021GG0225), the Central Public-interest Scientific Institution Basal Research Fund (No. 2023-YWF-ZYSQ-06), and the earmarked fund for CARS (CARS-34). There was no additional external funding received for this study. The funders had no role

in study design, data collection and analysis, decision to publish, or preparation of the manuscript.

## Grant Disclosures

The following grant information was disclosed by the authors:
Inner Mongolia Science and Technology Project: 2021GG0220, 2021GG0225.
Central Public-interest Scientific Institution Basal Research Fund: 2023-YWF-ZYSQ-06.
CARS: CARS-34.

## Competing Interests

The authors declare there are no competing interests.

## Author Contributions

- Zongyong Tong performed the experiments, analyzed the data, prepared figures and/or tables, and approved the final draft.
- Xianlin Dai performed the experiments, prepared figures and/or tables, and approved the final draft.
- Yu Wang analyzed the data, prepared figures and/or tables, and approved the final draft.
- Xianglin Li conceived and designed the experiments, authored or reviewed drafts of the article, and approved the final draft.
- Feng He conceived and designed the experiments, authored or reviewed drafts of the article, and approved the final draft.
- Guomei Yin conceived and designed the experiments, authored or reviewed drafts of the article, and approved the final draft.

## Animal Ethics

The following information was supplied relating to ethical approvals (*i.e.*, approving body and any reference numbers):

Experimental Animal Welfare and Ethical of Institute of Animal Science, Chinese Academy of Agricultural Sciences provided full approval for this research (IAS2021-247).

## Data Availability

The raw measurements are available in the Supplementary File.

## Supplemental Information

Supplemental information for this article can be found online at http://dx.doi.org/10.7717/peerj.17453#supplemental-information.

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
