# Peer review of "Analysis of rotational grazing management for sheep in mixed grassland"

_PeerJ, doi:10.7717/peerj.17453_

## Round 0.1 · original submission · Major Revisions

The reviewers and I all found this paper to be mostly well written and well organized. All four reviewers praised the robustness of the dataset and are eager to see the data published. However, there are some substantial questions raised by reviewers 2-4 concerning your experimental design and analytical approach. Therefore the determination at this time is that the paper will require a major revision prior to being considered further for acceptance to PeerJ.

Specifically the reviewers and I all feel that the analysis is somewhat lacking at this stage and in need of some changes/improvements.

Firstly, as reviewers 3 and 4 point out the description of the experimental units is lacking and needs to be better articulated. As reviewers 2 and 3 suggest the statistical models used are somewhat questionable. Given the multiple response factors among treatment groups the use of an uncorrected ANOVA is questionable. Use of MANOVA or a Generalized Linear Model is recommended. Additionally as reviewer 3 points out your line fitting and analysis with a product moment correlation is rather inappropriate, a regression analysis or other linear model would be preferred. Reviewer 4 suggests the addition of some meteorological data to better contextualize the study system.

Secondly, all four reviewers also point out several places where greater clarification is needed, with reviewer 3 providing some very specific editorial suggestions to help with the revision. As you revise, please pay careful attention to the concerns of Reviewer 3 as to the description of your treatment groups as this was somewhat nebulous in the current paper, and the study may indeed be better described as a mensurative study not a true experimental study due to lack of treatment replication. Discussion to this effect should also be added (Reviewer 4).

If you choose to make these revisions, the paper will likely be subject to additional peer-review by at least some of the original reviewers. Having said that, the reviews are mostly encouraging of the work and I too would strongly encourage you to make the suggested revisions and resubmit.

Sincerely,
Andy Gregory

Reviewer 1 ·

Basic reporting

Good context provided. Only a few places to check English language for revisions.

Experimental design

The research question here seems well defined and the methods are well described.

Validity of the findings

Data looks good and conclusions are well summarized and related back to the original scope.

Reviewer 2 ·

Basic reporting

In the manuscript, the authors have developed an artificial mixture grassland and implemented a rotational grazing system to investigate an effective sheep breeding model tailored for agro-pastoral transition zones. I very much appreciate the amount of work behind, several concerns should be addressed to improve the quality of publication.

While the manuscript is generally well-written, I recommend refining the language to better align with academic standards, particularly under "Fatty Acids" in the 'Materials & Methods' section.

It might be beneficial to incorporate the term "artificial grassland" for a more descriptive title.

When introducing the abbreviation 'ME' for metabolizable energy in the "plant nutrition" section, please provide its full form for clarity.

In Figure 3, please clarify the meanings of the labels 'a,' 'b,' and 'ab.'

Experimental design

Given that rotational grazing is a well-established topic, I suggest providing a background on existing grazing models and highlighting challenges associated with rotational grazing. This foundation will allow for a more robust discussion on how your approach addresses and potentially surpasses these traditional methods.

For the measurements of crude protein content and acid detergent fiber, referencing the methodologies or standards employed would enhance the manuscript's credibility.

While the manuscript asserts that the productivity of artificial mixed grassland surpasses that of natural grassland, the absence of a control group (sheep grazing on natural grassland) weakens this claim.

Validity of the findings

The manuscript delineates the concept and objectives of the artificial grassland. It would enhance the reader's understanding if guidelines for creating such grasslands were detailed, including species composition, selection criteria, and considerations for climate and seasonality. Additionally, it remains unclear whether the grassland was established by the authors or acquired from an external entity. If the latter, please specify the organization or company involved. Otherwise, referencing the methodologies or studies that influenced your decisions would be beneficial.

Also, considering the grassland's primary purpose is for sheep grazing, it would be pertinent to elaborate on whether the nutritional needs of the sheep were factored into its design and how these determinations were made for the artificial grassland.

The regions of the grassland, denoted as P1-6, would benefit from a visual representation indicating their respective grazing or resting periods.

Regarding Figure 4, the use of nonlinear curve fitting to depict the relationship between weight gain and total intake seems unjustified without an underlying mechanism.

While the manuscript emphasizes the benefits of increased crude protein for sheep, it's crucial to also address potential drawbacks. Protein needs vary across different growth stages of sheep, and imbalances can lead to health issues such as ammonia toxicity, urinary calculi, nitrate poisoning, or bloat from high alfalfa consumption. Additionally, other factors like minerals, vitamins, fiber, and environmental conditions significantly influence sheep's health. A comprehensive discussion on these aspects would enrich the 'Discussion' section."

Reviewer 3 ·

Basic reporting

I reviewed the manuscript Analysis of the rotation grazing breeding model for sheep in mixture grassland submitted by Tong et al. to PeerJ
I find the title odd and the wording confusing. We usually do not say rotation grazing breeding model. We simply use rotation grazing management and we also do not say mixture grassland but mixed species or multi-species grassland. I suggest to reword the title in the following way:

Analysis of rotational grazing management for sheep in mixed grassland

The study seems more of an observational type of analysis without any true replications of treatments and without replication of animal groups. The authors need to better justify the approach and why zones are important for this study.

Material & Methods lacks a description of imposed treatments and their replication. What do the zones stand for? Was each zone one paddock? We cannot infer based on which decision the zones were established and why these would be of interest for experimentation

I strongly recommend a professional fluent English speaker to polish the language

Experimental design

There is no experimental design underlying this study and the imposed treatments are not described. Thus, the present results are not unambiguous and the validity of the statistical analysis remains to be questioned. I also strongly recommend to better describe the rotational grazing management, with stocking rates, rotation lengths and number of rotations per year and zone. If the zone is the target of interest in your study then you need to provide a sophisticated argument why that is the case. Still the question remains why only one single group was studied without dividing the herd into replicated groups.

Validity of the findings

not possible to infer without information of treatments

Additional comments

Material & Methods lacks a description of imposed treatments and their replication. What do the zones stand for? Was each zone one paddock? We cannot infer based on which decision the zones were established and why these would be of interest for experimentation
L18: replace artificial mixture grassland into sown mixed grassland throughout the text
L89 what was the sowing rate in kg/ha, was the mixture divided by weight, how many seeds per m2 were sown?
L92 what is quadrat statistics? provide a reference
L96 rotational
What is the stocking density and stocking rate. What were the rotation lengths and how many rotations did you perform in total and per zone?
L97 replace hm2 by ha throughout the text
L103 how was it determined? To which residual was the grass grazed?
L110 standing aboveground herbage biomass it is 1 x 1 m or 1 m2
L111 pre- and post grazing not came in and exited
L112 forced-air oven? Which model
L117 and 118 pre- and post grazing
Why 49 x 5? What does the 5 stand for? It should be named apparent DMI as you present the average not from individual measurements
L121 near infrared reflectance spectroscopy (NIRS). What calibration is underlying the NIRS measurements and how are the typical SEC, SEP and SECV for ADF and CP?
L124 metabolisable energy (ME)
L148: there is no experimental design. What was the treatment factor and what were replicates? You cannot use ANOVA for multiple comparisons
Results
L152 throughout the text you often used “the results showed” which is uncommon. You need to be explicit and write, e.g. statistical analysis revealed a significant effect of xx on xx. For this, we would of course need to know the treatment factors and the design underlying this. However, in general please reword all “the results showed” throughout the text
L154-156 refers to MM part
L164-166 belongs to MM
Conclusion is a replication of results. Avoid any numbers and statistical measures in the conclusion

Reviewer 4 ·

Basic reporting

The presented paper titled "Analysis of the Rotation Grazing Breeding Model for Sheep in Mixture Grassland" contains important information on the impact of land use on sheep grazing and their behavior parameters. The publication contains important information that is not widely available in the literature data. The paper is generally well constructed. Contains appropriately listed chapters and subchapters, however, some individual content requires improvement.

Experimental design

In the M&M section, if chemical methods are used (e.g. fatty acids determination), all metrological parameters of the method should be added (detection limit, quantification limit, precision, recovery, uncertainty, etc.).

Validity of the findings

The discussion requires extension because it contains only 10 citations, which is quite poor in terms of the conducted research. The discussion could be expanded to include the impact of feeding on other four-hoofed animals to provide a broader context.

Moreover, the conclusions need to be reworded because they do not constitute an appropriate summary of the research.

The description of the results should indicate how many male and female individuals there were and whether the results differ statistically significantly for these groups within the parameters examined.
Please standardize the units of the results - especially in fatty acids, change permille to mg/kg.

Additional comments

no comment

---

## Round 0.2 · Minor Revisions

Thank you for your patience as we reviewed your revised and resubmitted documents. Your attention to detail in addressing the reviewer comments and concerns was much appreciated.

Both of the reviewers and myself agree that the addition methodological detail and the change in the statistical model used greatly improved the manuscript, and we now find the paper to be technically sound and valid.

However, both reviewers also feel that the manuscript could use some additional copy editing for language, consistency, and clarity. Therefore, my recommendation at this time that an additional minor revision is still warranted.

Reviewer 3 provides some specific recommendations with line number references for where some specific grammatical oddities occur and makes some suggestions on how to improve clarity an consistency. However, I would encourage a comprehensive editorial review for clarity and consistency in how you describe your study system and results.

Thank you for your patience and continued efforts. We look forward to receiving a revised version in the near future.

Sincerely,
Andy Gregory

**Language Note:** The Academic Editor has identified that the English language must be improved. PeerJ can provide language editing services - please contact us at [email protected] for pricing (be sure to provide your manuscript number and title). Alternatively, you should make your own arrangements to improve the language quality and provide details in your response letter. – PeerJ Staff

Reviewer 2 ·

Basic reporting

The revised manuscript exhibits commendable improvements, notably in clarity, data presentation, and addressing prior concerns. The refined language and grammar enhance readability. To further enrich the manuscript, briefly discuss potential future research directions in the conclusion. With minor revisions, this manuscript is ready for publication.

Experimental design

N/A

Validity of the findings

N/A

Reviewer 3 ·

Basic reporting

My comments refer to the revised manuscript

Please add scientific names at first mention of a species, e.g. in your intro or the MM section

e.g. we read in 60-64:
It is generally composed of perennial legume (alfalfa, clover, etc.) and grass (ryegrass, smooth brome, Leymus chinensis, etc.) species, combinations of these species in different ecological niches could make full use of natural environmental resources, and achieve nutritional balance.

L64: The typical example is the clover-ryegrass grassland in New Zealand. why is that? it is not only New Zealand

or in L136 ff in the MM section where the species composition is described in detail

I doubt that the English language has been checked by a native speaker as was written in the response letter. Here are my indications why the language is still very unsatisfying

L72-76:"The existing grazing models mainly include continuous grazing and rotational grazing, continuous grazing in which livestock have unrestricted access to the pasture area often results in overgrazing and pasture degradation. The rotational grazing system for moving livestock from paddock to paddock are based on available forage, paddock size and livestock growth goals. "
L137 ff
The mixed sowing grassland was planted in September 2016, and the sowing mode was mixed legume and grass, in which the legume was alfalfa, the sowing rate was 30 kg/ha, accounting for 75%, and the grasses were tall fescue and orchardgrass, the sowing rate was 5 kg/ha respectively, accounting for 12.5% in weight.

L153 ff
The rotation grazing test was conducted on May 6, 2021, the stocking rate was 30 ewes/ha, the rotation time in per paddock was 5 days, rotation length per circle was 30 days, and 5 rotation circles were conducted per year.
L440-441:
"Deceased aboveground biomass after grazing in each paddock and DMI for every sheep were presented in Table S1." it should be are given in Table S1 not were
L669 ff
To prevent bloating from high alfalfa consumption, sheep were adaptive trained with feeding them grass hay and concentrate, and gradually extend the grazing time (Wang et al. 2023; Wang et al. 2012). And to prevent various diseases caused by nutritional imbalance, therefore, energy feeds, such as corn, are needed to achieve energy and protein balance.

and other areas which I will not highlight
L664 nutritional
L666 delete and harvesting
L683 what is the expected weight gain? Reference
L713 birdsfoot
L715 pastures
L728 acid

Experimental design

it is now much better explained. The animals are treated as replicate. This is a justified approach since the study is not focussing on behaviour but the performance

Validity of the findings

I think the wording is still very unusual. Phrases like the "existing grazing models..." in L74 are usually not used.
It is also not planted but sown (L137) and there are no "sowing modes"

Additional comments

I have major concerns with the language and have requested in my previous review that the paper should be checked. This might have happened but I doubt that this was done thoroughly. I recommend another language check. In addition I recommend the check by a renown grassland scientist who is aware of the common terminology that is usually used in scientific papers in this respect. Some starting points can be found in Allen et al. (2011) grassland terminology. The authors have introduced some literature to the reference list. Maybe a close look into the descriptions of this papers will help to find the appropriate wording.

---

## Round 0.3 · accepted · Accept

This MS has gone through a good review process and the co-authors have responded well to the suggestions from the expert reviewers over two rounds of review. They have improved the language as well, and have provided detailed responses along with their revisions. At this point, I believe that this paper is ready for publication in PeerJ.